# Identification and Characterization of CYP6 Family Genes from the Oriental Fruit Moth (*Grapholita molesta*) and Their Responses to Insecticides

**DOI:** 10.3390/insects13030300

**Published:** 2022-03-17

**Authors:** Hui Han, Yanyu Yang, Jun Hu, Yuanxin Wang, Zhiguo Zhao, Ruiyan Ma, Lingling Gao, Yanqiong Guo

**Affiliations:** 1College of Plant Protection, Shanxi Agricultural University, Jinzhong 030801, China; hanhui907@126.com (H.H.); yangyanyu1999@163.com (Y.Y.); hujun.yx@163.com (J.H.); wangyuanx1992@163.com (Y.W.); nice2me@126.com (Z.Z.); maruiyan2019@163.com (R.M.); 2CSIRO Agriculture & Food, Private Bag 5, Wembley, Perth, WA 6913, Australia

**Keywords:** *Grapholita molesta*, cytochrome P450 monooxygenases, expression, malathion, deltamethrin, chlorantraniliprole

## Abstract

**Simple Summary:**

Eight CYP6 family genes were identified in *Grapholita molesta* (Busck). The expression of individual gene members differed between the developmental stages and insect tissues. High expression was found in third/fourth-instar larvae and in the midgut and Malpighian tubules. The response patterns of the genes exhibited diverse response patterns to the three representative insecticides were diverse.

**Abstract:**

Cytochrome P450 (CYP) monooxygenases comprise a superfamily of proteins that detoxify xenobiotics and plant secondary metabolites in insects. The CYP6 family is unique to the class Insecta, and its members participate in the metabolism of exogenous substances. In this study, we sequenced and characterized the full-length cDNAs of eight CYP6 family genes from *Grapholita molesta* (Busck), a global pest of pome fruits. P450 genes with the exception of *CYP6AN35*, which was most highly expressed in adults, consistently showed high expression in third- or fourth-instar larvae. The analysis of different tissues of adults showed that most of these genes were predominantly expressed in the midgut, Malpighian tubules, and/or fat body. The expression of these eight CYP6 genes was differentially affected by three representative insecticides: malathion (organophosphate), deltamethrin (pyrethroid), and chlorantraniliprole (carbamate). All eight CYP6 genes responded to malathion treatment. Only three CYP6 genes were highly expressed in deltamethrin-treated individuals. Chlorantraniliprole treatment exerted weak effects on gene expression. Interestingly, *CYP6AN35* was a highly expression level in the adult head and its expression was induced by all three insecticides. *CYP6AN35* may be a key gene in the metabolism of insecticides. This study provides a fundamental understanding of the functions of the CYP6 gene family in insecticide metabolism in *G. molesta*.

## 1. Introduction

The global fruit-borne pest *Grapholita molesta* (Busck) (Lepidoptera: Tortricidae) causes serious harm to horticultural crops such as pear and peach. Specifically, this pest affects the yield and quality of crops by drilling into fruits and sprouts [1,2,3]. Due to the increases in the orchard area and climate change, the population of *G. molesta* has increased sharply and the damage to fruit trees has become increasingly severe [3,4,5]. The chemical control of this pest remains the dominant strategy, but the repeated use of large amounts of chemical pesticides has led to the development of resistance in this insect [6]. In the United States, Canada and other countries, *G. molesta* has become resistant to the major modes of insecticides [7,8,9], such as organophosphates, carbamates, pyrethroids, and other insecticides [10].

Many studies have shown that the resistance mechanisms include the increased metabolic detoxification of insecticides and target site insensitivity [11]. Metabolic resistance is mainly caused by elevated activities of detoxifying enzymes, such as cytochrome P450 (CYP), glutathione S-transferases (GSTs), and carboxylesterases (CarEs) [12]. Cytochrome P450 are hemoproteins that act as terminal oxidases in monooxygenase systems, and they compose a large superfamily of proteins that are found in virtually all living organisms [13]. CYP plays a major role in phase I (biotransformation) detoxification [14]. The number of CYP genes varies greatly among different insect species, for example, 91 genes have been identified in *Drosophila melanogaster* [15], and 158, 78, and 146 genes have been found in *Aedes aegypti* [16], *Bombyx mori* [17], and *Cydia pomonella* [18], respectively. The different CYP genes are primarily responsible for the various functions of the P450 superfamily. More than 1000 genes have been identified to date, and these belong to the CYP4, CYP6, CYP9, CYP12, CYP15, and other CYP families [19]. Among these families, the CYP4, CYP6, CYP9, and CYP12 families have been linked to insecticide resistance [12,20]. The CYP6 family is unique to the class Insecta, and many studies have shown that its members participate in metabolizing exogenous and plant secondary substances [21]. A study on *Locusta migratoria* suggested that the CYP6 family plays a key role in the susceptibility of this insect to both carbamates and pyrethroids [22]. By knocking out the CYP6AE gene cluster, Huidong Wang et al. revealed a role of its members in the detoxification of phytochemicals and insecticides by *Helicoverpa armigera* [23]. Furthermore, studies on *Bemisia tabaci* showed that *CYP6CM1* is associated with the metabolism of the neonicotinoid class of insecticides [24].

Gene sequencing technology provides the opportunity to identify more P450 genes. For example, with this technology, 120 CYP genes were identified in *Pardosa pseudoannulata* [25]. Transcriptome analysis of the fat body of *Procecidochares utilis* revealed many unigenes involved in detoxification, including 50 unigenes of putative CYPs (P450s) in this organism [26]. We have previously reported 77 unigenes of putative CYPs in *G. molesta*, which included 13 unigenes in the CYP6 family [27], the biochemical functions of which are largely associated with the metabolism of xenobiotics [21]. However, little is known about the roles of individual members of the CYP6 family in insecticide detoxification in *G. molesta*.

In this study, eight CYP6 genes were identified through bioinformatics analysis, and their molecular structures were predicted. Their expression at different developmental stages and three tissue specificity were analyzed. In addition, the expression levels of CYP6 genes in *G. molesta* in response to three representative insecticides (malathion, deltamethrin, and chlorantraniliprole) were determined using a regression equation what allowed the prediction of the target insecticide of these CYP genes. This approach provided a fundamental understanding of the functions of the CYP6 gene family in insecticide metabolism and thus, providing a theoretical basis for the exploration of new pest control management for *G. molesta*, such as novel and durable pesticides.

## 2. Materials and Methods

### 2.1. Insect Samples

The sampled *G. molesta* individuals sampled were obtained from Taigu, Shanxi Province, China, and originated from a population that had been reared indoors with an artificial diet without exposure to any insecticide for more than five years. The growth environment had a temperature of 26 ± 1 °C, a photoperiod of 15:9 h (light:dark), a relative humidity of 60–80%, and a light intensity of 4800 lx [28,29]. To ensure the precision of the insect sample, we selected healthy adults of uniform size that were all within 24 h of emergence.

*G. molesta* individuals at different developmental stages, including three adults, five pupae, five fifth-instar larvae, five fourth-instar larvae, ten third-instar larvae, thirty second-instar larvae, fifty first-instar larvae, and a hundred eggs were collected, and three biological replicates were established. The head, foregut, midgut, hindgut, fat body, Malpighian tubules, spermary, and ovary were harvested from healthy adults of *G. molesta* within 24 h of emergence, with sixty adults comprising a replicate and three biological repetitions were established.

RNA was extracted from the obtained samples and used as a template for reverse transcription to cDNA. Quantitative PCR (qPCR) was used to quantify the initial expression levels of different genes at different developmental stages and in different tissues of *G. molesta* adults.

### 2.2. Bioinformatics Analysis

The nucleic acid sequences of eight CYP6 genes were obtained from the transcriptome of *G. molesta* [27]. ORFfinder (http://www.ncbi.nlm.nih.gov/orffinder, 25 January 2022) was employed to screen nucleic acid sequences with complete ORFs, and the sequences were submitted to the Cytochrome P450 International Nomenclature Committee for naming. NCBI BLASTp was used to align the protein sequences, and the ExPASy ProtParam tool (http://web.expasy.org/protparam/, 25 January 2022) was used for the analysis of parameters such as protein molecular weight, amino acid composition, acid-based amino acid number, and total average hydrophilicity. Signal peptides were examined with the SignalP4.1 server (http://www.cbs.dtu.-dk/services/SignalP/, 25 January 2022). ProtScale (http://web.expasy.org/protscale/, 25 January 2022) was applied for the prediction of protein hydrophobicity and AlphaFold2 (https://colab.research.google.com/github/sokrypton/ColabFold/blob/main/AlphaFold2.ipynb, 25 February 2022) was applied for the prediction and simulation of the protein tertiary structure. A MEGA 7.0 cluster analysis of amino acid sequences was performed, and forty-three P450 genes were chosen from *B. mori*. The neighbor-joining (NJ) method with 1000 bootstrap replicates was used to construct a phylogenetic tree in MEGA 7.0 [30].

### 2.3. Induction of Expression by Three Kinds of Insecticides

Three insecticides including malathion, deltamethrin, and chlorantraniliprole were used and obtained from AccuStandard (New Haven, CT, USA). Each insecticide was dissolved with acetone. Concentrations are: malathion, LC_10_ (1.33 μg mL^−1^), LC_30_ (2.02 μg mL^−1^), and LC_50_ (3.08 μg mL^−1^); deltamethrin, LC_10_ (1.04 μg mL^−1^), LC_30_ (5.98 μg mL^−1^), and LC_50_ (23.52 μg mL^−1^); and chlorantraniliprole, LC_10_ (32.36 μg mL^−1^), LC_30_ (66.07 μg mL^−1^), and LC_50_ (130.78 μg mL^−1^) [31]. The insecticides or acetone (control) were applied to newly emerging adults of *G. molesta*. After 24 h, the surviving adults were quickly frozen in liquid nitrogen for subsequent analysis.

We isolated total RNA from the adults using RNAiso Plus (Takara, Dalian, China). The RNA concentration and purity were estimated by using a Nanodrop 2000 spectrophotometer (Thermo Fisher Scientific, Waltham, MA, USA) with 1 μL of a 50-fold dilution of the RNA. RNA integrity was detected by using 2–5 μL of each sample for agarose gel electrophoresis.

First-strand cDNA was synthesized by following the instructions for the Takara Reverse Transcriptase Kit (Takara, Dalian, China). The reaction product was stored at −20 °C. *β-actin* of *G. molesta* was used as an internal reference gene [32]. The qPCR primers were designed using Primer Premier 5.0 software (Appendix A) and synthesized by Sangon Biotech (Shanghai, China). The PCR primers used are detailed in the Appendix A.

### 2.4. Statistical Analysis

After the reactions, the ABI 7500 software data were exported to Microsoft Excel (Microsoft, WA, USA) and analyzed using the 2^−ΔΔCT^ method [33] to determine the expression levels of CYPs in *G. molesta* exposed to different concentrations.

One-way ANOVA followed by Tukey’s honest significant difference (HSD) test at *p* < 0.05 were performed to determine whether the normalized expression level of the eight CYP6 genes differed significantly among developmental stages, tissues, and insecticide treatments. SPSS 22.0 software (IBM, Armonk, NY, USA) was used. All histograms were constructed using SigmaPlot 12.0 (Systat Software, Inc., San Jose, CA, USA).

## 3. Results

### 3.1. Identification and Characteristics of CYP6 Gene Members in G. molesta

Eight CYP6 genes were identified from the transcriptomic data of *G. molesta* (Appendix A). These genes encode putative proteins consisting of 498–530 amino acid residues, with relative molecular weight of 58–61 kDa and isoelectric points (pIs) of 6.55–8.84. The instability coefficients were less than 40% and *CYP6AB112*-*CYP6AB115* does not contain signal peptide (Appendix A). Multiple-sequence alignment of the eight genes revealed common CYP signature sequences (Appendix A). All genes contain the characteristic sequences of the heme-binding region, i.e., the ‘FXXGXXXCXG/A’ sequence in the meander, ‘PXXFXPXXF’, and the P450 oxygen binding sequence, i.e., ‘A/GGXD/ETT/S’. The newly obtained protein sequences were aligned with protein sequences of known functions, and the structures and functions of the new protein sequences were predicted. The proteins encoded by the eight CYP6 genes had similar tertiary structures (Figure 1). The purple region was the heme-binding region, which is the most conserved structural center of CYP6, with a noncovalently bound heme and the surrounding highly conserved cysteine sequence ‘FXXGXXXCXG/A’.

### 3.2. Phylogenetic Analysis

A phylogenetic analysis of the *G. molesta* CYP6 proteins along with those known in *B. mori* was performed [34]. All eight genes belonged to the CYP3 clan and six of them and the *B. mori* CYP6 subfamily formed a cluster, whereas *CYP6B74* and *CYP6AB117*, respectively, formed a cluster with other genes of the CYP6 subfamily in *B. mori* (Figure 2).

### 3.3. Developmental Stage and Tissue Specificity of P450 Gene Expression

To gain insight into the functional differences, the relative mRNA expression levels of the eight genes in different developmental stages, including eggs, first- to fifth-instar larvae, pupae, and adults, were examined. As shown in Figure 3, *CYP6AB112*, *CYP6AB114,* and *CYP6B74* expression increased as development progressed until it peaked in third-instar larvae, and *CYP6B74* peaked in fourth-instar larvae. *CYP6AB113* and *CYP6AB117* exhibited the same expression pattern, which consisted of gradually increasing before the fourth-instar stage, peaking at the third-instar stage, and showing a small increase from the fourth- to fifth-instar larval stage. *CYP6AN35* and *CYP6AB115* were found at high expression levels in adults. *CYP6AN35* expression increased gradually, and reached the highest level in adults, and *CYP6AB115* expression peaked in the third-instar larval stage. *CYP6AB116* showed little change in expression across the different stages and exhibited significantly higher expression in the egg stage than at the other stages.

The expression patterns of the eight CYP6 genes among different tissues, including head, foregut, midgut, hindgut, fat body, Malpighian tubules, spermary, and ovary, were also examined. The results showed that the highest expression of *CYP6AN35* was found in the head and that the other seven CYP6 genes were highly expressed in the midgut. In addition, the expression of *CYP6AB112, CYP6AB114, CYP6AB115*, and *CYP6AB117* was widely distributed in Malpighian tubule, and *CYP6AB115* expression was significantly highest in Malpighian tubule than in other tissues. The expression of *CYP6AB115* was higher in the ovary than in other tissues. *CYP6AB113* and *CYPAB116* exhibited high expression in fat bodies, and the expression of *CYP6AB112*, *CYP6AB113,* and *CYP6AB116* was higher in the spermary (Figure 4).

### 3.4. Response of CYP Genes Expression to Insecticide Exposure

Three common insecticides, namely, malathion, deltamethrin, and chlorantraniliprole were selected to evaluate their possible effects on the expression of eight CYP6 genes in this study. The transcriptional changes in the eight CYP6 genes in adult moths exposed to the insecticides were determined by RT-qPCR. Malathion treatment increased the *CYP6AB113*, *CYP6AB115,* and *CYP6AN35* expression levels in a concentration-dependent manners. In contrast, the expression level of the *CYP6AB114* gene decreased and remained lower than the control level. With increases in the malathion concentration, the expression of *CYP6AB116* and *CYP6AB117* showed fluctuating trends, with peak expression at LC_30_, although the expression changes of these genes were small. No significant differences in *CYP6AB112* expression were observed, and the expression of *CYP6B74* was a high expression only with the LC_50_ treatment (Figure 5).

The expression of CYP6 genes in *G. molesta* treated with different concentrations of deltamethrin was analyzed (Figure 6). Treatment with deltamethrin rapidly decreased the expression of *CYP6AB112, CYP6AB114,* and *CYP6AB116* after 24 h exposure. With increases in the insecticide concentration, the expression of these genes did not increase significantly and remained markedly lower than that of the control after 24 h exposure. These results demonstrated that the expression of these genes maybe be inhibited by deltamethrin within a period of time. *CYP6AB115*, *CYP6AN35,* and *CYP6B74* showed the highest expression levels with the LC_30_ concentration. Compared with the levels obtained with the control treatment, the *CYP6AN35* and *CYP6B74* expression levels only increased 2.5–3.0-fold after deltamethrin treatment. In addition, the expression of *CYP6AB113* and *CYP6AB117* did not significantly differ among the treatments with different deltamethrin concentrations.

Treatment with different concentrations of chlorantraniliprole (Figure 7) decreased the expression levels of *CYP6AB114, CYP6AB116, CYP6AB117,* and *CYP6B74*, and the fold changes in expression showed subtle differences among the treatments. Among these genes, *CYP6AB116* and *CYP6AB117* exhibited the same expression pattern. The expression of *CYP6AB112* and *CYP6AB113* peaked at LC_10_, reaching 1.8–4.0-fold that obtained with the control treatment, and then decreased with increases in the chlorantraniliprole concentration. Compared with the levels obtained with the control treatment, *CYP6AB115* expression decreased with the lower chlorantraniliprole concentrations and increased with the highest concentration of chlorantraniliprole. The *CYP6AN35* expression levels were highest with the LC_30_ treatment, and these levels were higher than those obtained with the control treatment.

## 4. Discussion

CYP monooxygenases constitute a common superfamily in insects. Studies have shown that these enzymes can participate in the detoxification of xenobiotics and plant secondary metabolites in insects [35,36]. The CYP6 family is unique to the class Insecta, and the biochemical functions of its members are predominantly associated with the metabolism of xenobiotics [37]. Many studies have contributed to our understanding of the relationships between the molecular characteristics and metabolic detoxification functions of CYP genes [38]. We previously performed transcriptome sequencing of *G. molesta* and obtained 77 P450 unigenes [27]. In the present study, eight CYP6 family genes containing conserved sequences of CYPs were identified by transcriptomic and bioinformatics analyses. These genes were named by the Cytochrome P450 International Nomenclature Committee as follows: *CYP6AB112, CYP6AB113, CYP6AB114, CYP6AB115, CYP6AB116, CYP6AB117, CYP6AN35,* and *CYP6B74*. According to their molecular characteristics, these eight genes had typical CYP structures and shared the closest relationships with genes from the Lepidoptera species *C. pomonella, Manduca sexta, B. mori*, *Amyelois transitella, Heortia vitessoides, Ostrinia furnacalis,* and *Cnaphalocrocis medinalis* [39,40,41]. These results provide an important basis for the construction of three-dimensional structures that are crucial for understanding protein function at the molecular level. Starting from the amino acid sequences of the interacting proteins, both the stoichiometry and the overall structure of the complex were predicted by homology modeling which could help to facilitate further study of the functions of the genes at the protein level [42,43].

In *G. molesta,* the expression of CYP6 gene members differed among the different developmental stages, which suggested that these genes may be involved in different metabolic processes. Seven of these eight CYP6 genes exhibited low expression in eggs, only *CYP6AB116* which may participate in worm survival and egg development presented slightly high expression. For example, in *Schistosoma mansoni*, a high expression level of *CYP3050A1* is found in eggs and double-stranded RNA (dsRNA) silencing results in worm death [44]. The expression of the CYP6AB genes was significantly higher during the third-instar larval stage of *G. molesta* and the highest expression of *CYP6B74* was found at the fourth-instar larvae. These genes may be involved in the metabolism of foreign substances because, third and fourth instars are the stages involving the large-scale feeding of *G. molesta*, which possibly require more enzymes to metabolize exogenous substances. In *Bradysia odoriphaga*, *CYP3356A1* presented the highest expression level in fourth-instar larvae and is related to the metabolism of exogenous substances [45]. *CYP6AN35* and *CYP6AB115*, which are highly expressed in adults, may participate in the detection and metabolism of foreign substances. In *L. migratoria*, high expression levels of *LmCYP6HQ1* were found in adults, through RNAi knockdown of *LmCYP6HQ1*, which suggested that the expression of genes enhances the locust’s ability to detoxify insecticides [22]. *CYP6BQ23* in *Meligethes aeneus* is also significantly and highly overexpressed (∼900-fold) in adults of pyrethroid-resistant strains compared with susceptible strains, and is the primary mechanism conferring pyrethroid resistance [46].

In *G. molesta,* the expression intensity of CYP6 members was tissue specific. *CYP6AN35* was most highly expressed in the head and may be related to the recognition and metabolism of toxics substances. In *Plutella xylostella* (Linnaeus), high expression of *CYP4BN6* in the head of *Tribolium castaneum* has been suggested to be associated with insecticide susceptibility [47]. Studies have shown that certain CYPs with a high expression level in the insect head participate in olfaction or detoxification [48]. Among the eight CYP6 genes analyzed in this study, all of the genes with the exception of *CYP6AN35* exhibited higher levels to varying degrees in the Malpighian tubules and fat body. Similar results have been obtained in many other studies on P450s [49]. Most of the P450 enzymes functioning as metabolic detoxification are highly expressed in the midgut, fat body, and Malpighian tubules. Midgut and fat body are major tissues involved in the detoxification of allelochemicals [50,51]. In addition to maintaining the salt and water balance, the Malpighian tubules play key roles in the detoxification and excretion of xenobiotics [52]. *CYP6AB56* is highly expressed in the fat body of *Mamestra brassicae* Linnaeus and RNA interference followed by a deltamethrin bioassay showed that it plays an important role in deltamethrin detoxification [53]. These findings indicate that in *G. molesta*, CYP6 genes may be involved in metabolic detoxification. The diversity of the gene distribution in the insect body indicates that the genes may be involved in different metabolic activities [20,54].

Many results have demonstrated that CYPs participate in the insecticides’ metabolism. To clarify which CYP genes are involved in the metabolism of insecticide, we investigated the effect of the three insecticides on CYP gene expression. Gene exposure to endogenous and exogenous substances was induced differently at different times. Zhang et al. found that the expression of many P450 genes reached the highest level at 24 h after induction [55]. Our study considered gene expression changes after 24 h exposure, and the result indicated a complex relationship between CYP genes and insecticide treatments. Among these eight CYP6 genes, *CYP6AB112* was only highly expressed after chlorantraniliprole treatment, was uninfluenced by malathion, and was downregulated by deltamethrin. This result preliminary certifies that this gene may play roles in the chlorantraniliprole metabolism. Similar results have been reported for *CYP6FD1* in *L. migratoria*, which was inhibited by deltamethrin treatment and no significant increase in mortality was found after RNAi knockdown [37]. We also found that *CYP6AB115* and *CYP6AN35* expression increased after treatment with the three insecticides, and these two genes may play an important role in metabolic detoxification in *G. molesta.* The same result for *B. odoriphaga CYP3356A1* revealed its association with the metabolism of three xenobiotics, imidacloprid, thiamethoxam, and beta-cypermethrin through RNAi [45]. The expression of *CYP6AB114* showed the same decreasing pattern in response to different insecticides after 24 h exposure. Treatment with different concentrations resulted in lower expression levels of this gene than those obtained with the control treatment, and this gene may thus play a role in the adaptive response to external stimuli. Decreased expression of P450 is a homeostatic or adaptive response, and suppressed in response to pathophysiological signals and various exogenous or endogenous compounds [56,57]. *CYP6AB116* and *CYP6AB117* expression showed little change across the different concentrations of these three insecticides, which suggested little involvement of these genes in resistance to these chemicals. *CYP6B74* expression was induced by malathion and deltamethrin, and may participate in the metabolism of these two insecticides. A study on the CYP6B family showed that *CYP6B1* metabolizes organophosphorus diazinon, and *CYP6B8* metabolizes both diazinon and the pyrethroid cypermethrin [58]. *CYP6BB2* upregulation can be observed in pyrethroid-resistant populations and is associated with pyrethroid resistance [59].

*CYP6AN35* exhibited a special expression pattern in the head of *G. molesta* adults, and our results showed that the expression of this gene tended to be induced by three insecticides, which indicates that this gene might be a key gene for further exploration. For example, the upregulation of *CYP6G1* confers resistance to DDT [60] and high expression of *CYP9AQ2* plays an important role in deltamethrin detoxification in *L. migratoria* [61]. An analysis of insect detoxification enzymes has been proposed as a means for identifying the main enzymes involved in insecticide detoxification [62]. Determining the genes that participated in the insecticide metabolism remains difficult in our study due to the diversity, rapid evolution, and lack of information about the substrate specificity of cytochrome P450 enzymes [63]. Our study clearly demonstrates that the expression of multiple P450 genes was significantly enhanced after treatment with different insecticides. The diversity of cytochrome P450 genes in organisms constitutes an adaptation of insects to their plant hosts and environmental change. This result provided possible target genes for the exploration of the metabolic detoxification function in *G. molesta* and corroborated the finding that *G. molesta* cytochrome P450 responds by changing its expression in the face of pesticide stress. Our findings revealed the functional importance of cytochrome P450 genes in response to insecticide exposures, the detoxification of insecticides, the selection of insecticides for *G. molesta* control, and the evolution of insecticide resistance in *G. molesta*.

## 5. Conclusions

In this study, eight cytochrome P450 genes were identified and found to exhibit various expression levels at different developmental stages and in different tissues. Most of these eight genes were highly expressed in third- or fourth-instar larvae and were distributed in detoxification-related tissues, such as the midgut, Malpighian tubules, and/or fat body in adults. Interestingly, *CYP6AN35* was found at a high expression level in the adult head and its expression was induced by all three insecticides. *CYP6AN35* may be a key gene in the metabolism of insecticides.

## Figures and Tables

**Figure 1 insects-13-00300-f001:**
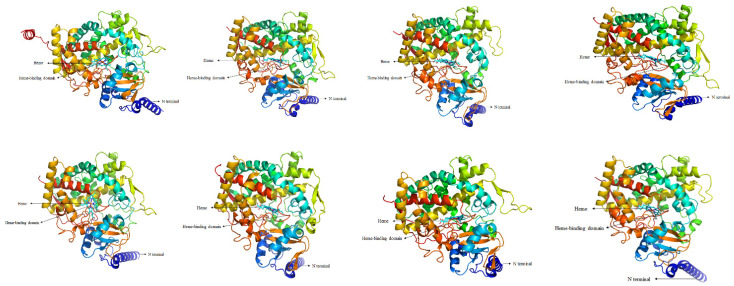
Predicted tertiary structures of eight CYP6 genes from *G. molesta*. Image colored by rainbow N-C terminus. (Appendix A).

**Figure 2 insects-13-00300-f002:**
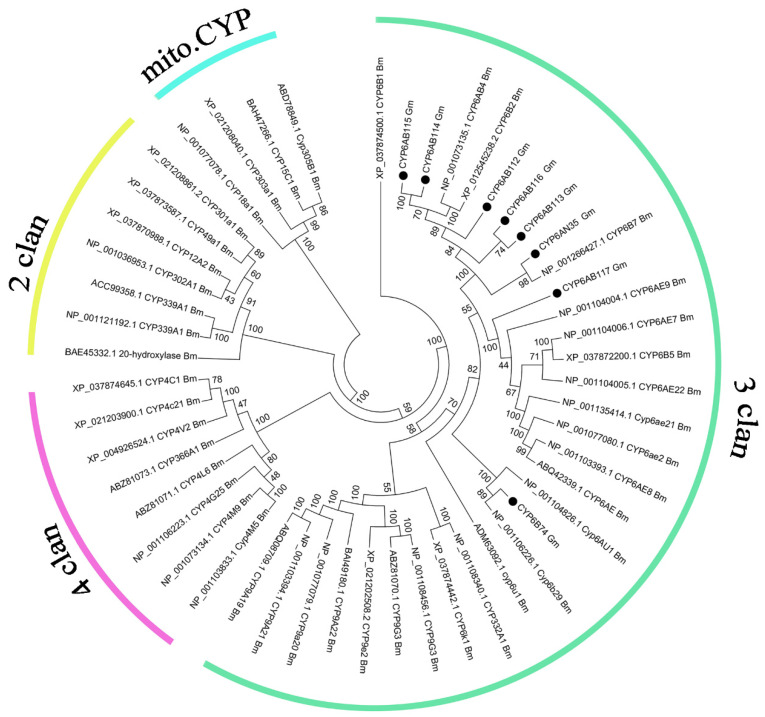
Phylogenetic analysis of eight CYP6 genes in *G. molesta* and *B. mori* CYPs. Forty-three CYP sequences from *B. mori* were used to construct this phylogenetic tree. The eight *G. molesta* CYPs are indicated by “●”.

**Figure 3 insects-13-00300-f003:**
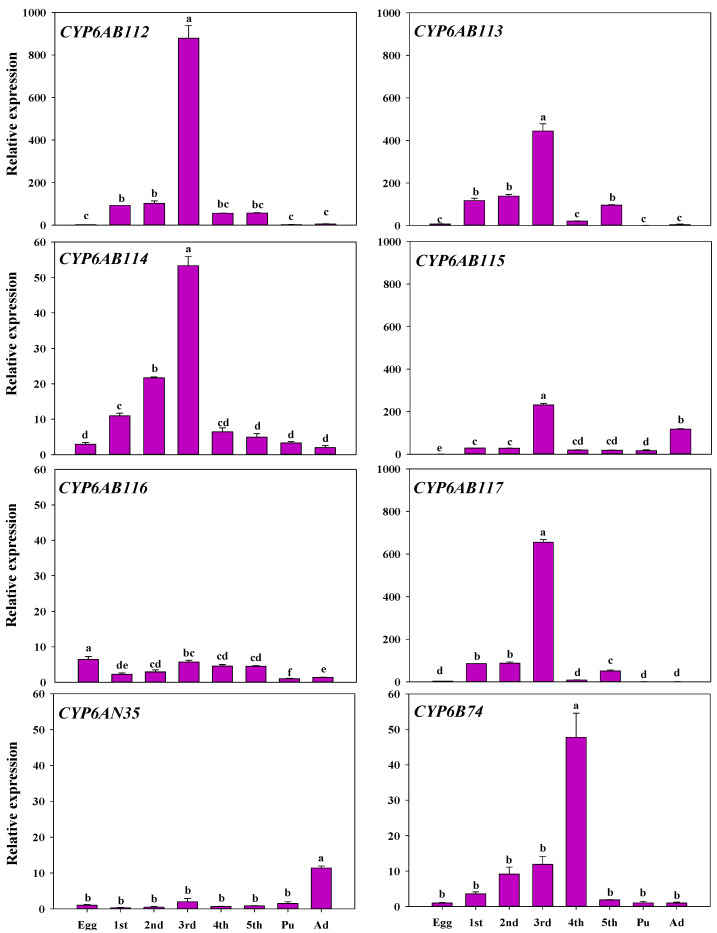
Expression of eight CYP6 genes at different developmental stages of *G. molesta* Note: egg (Egg), first-instar larvae (1st), second-instar larvae (2nd), third-instar larvae (3rd), fourth-instar larvae (4th), fifth-instar larvae (5th), pupae (Pu), and adults (Ad). Each treatment included three biological replicates. The data in the bar charts are presented as the means ± SEs. Different letters indicate significant differences at the *p* < 0.05 level, as determined by ANOVA and Tukey’s HSD test.

**Figure 4 insects-13-00300-f004:**
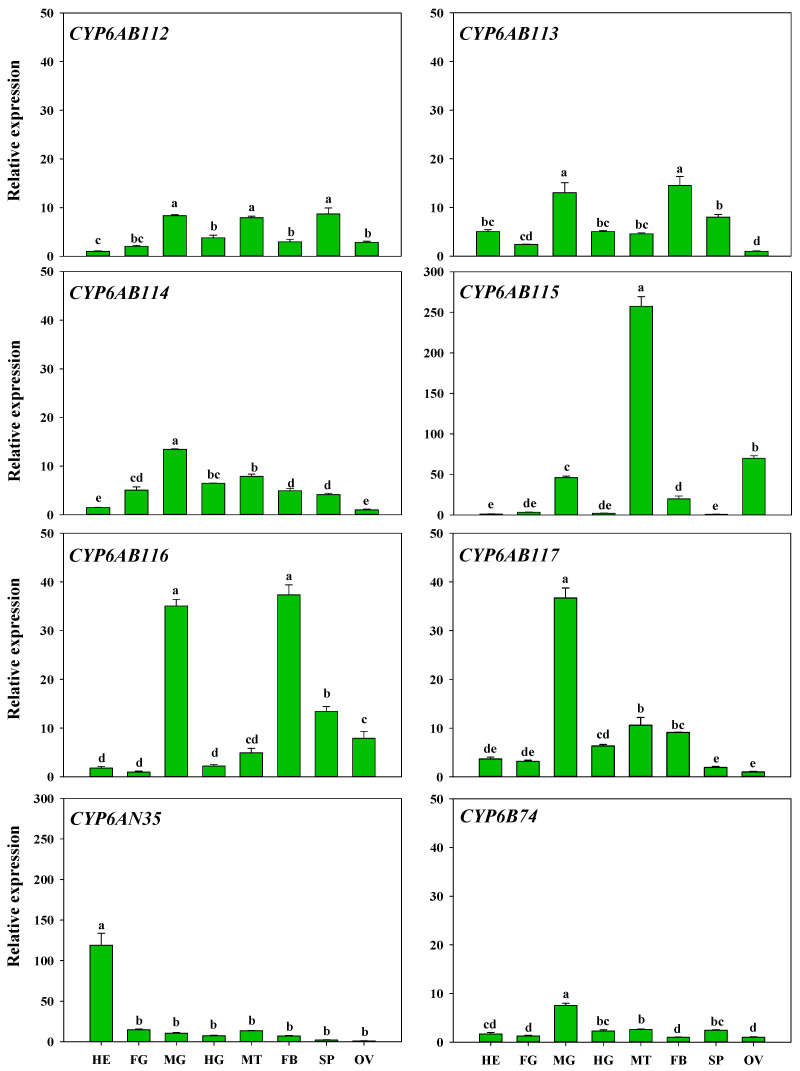
Expression patterns of eight CYP6 genes in different tissues of *G. molesta*. Note: head (HE), foregut (FG), midgut (MG), hindgut (HG), Malpighian tubule (MT), fat body (FB), spermary (SP), and ovary (OV). Each treatment included three biological replicates. The data in the bar charts are presented as the means ± SEs. Different letters indicate significant differences at the *p* < 0.05 level, as determined by ANOVA and Tukey’s HSD test.

**Figure 5 insects-13-00300-f005:**
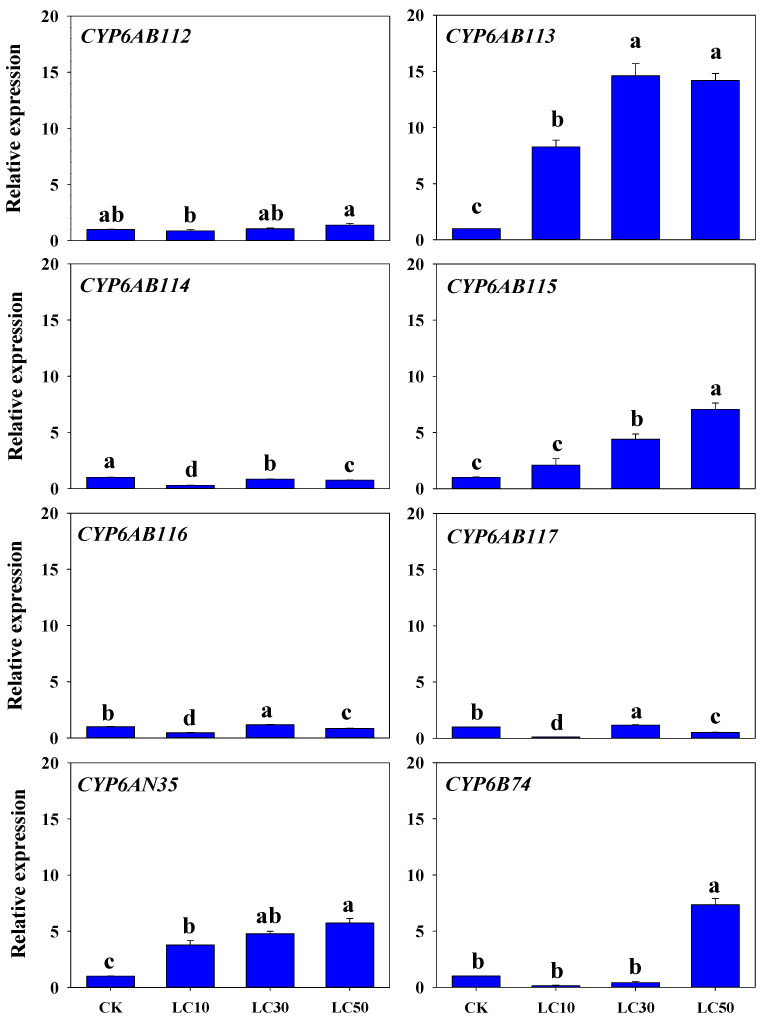
Relative expression of CYP6 genes in *G. molesta* treated with different concentrations of malathion. Note: control (CK), lethal concentration 10% (LC_10_), lethal concentration 30% (LC_30_), and lethal concentration 50% (LC_50_). Each treatment included three biological replicates. The data in the bar charts are presented as the means ± SEs. Different letters indicate significant differences at the *p* < 0.05 level, as determined by ANOVA and Tukey’s HSD test.

**Figure 6 insects-13-00300-f006:**
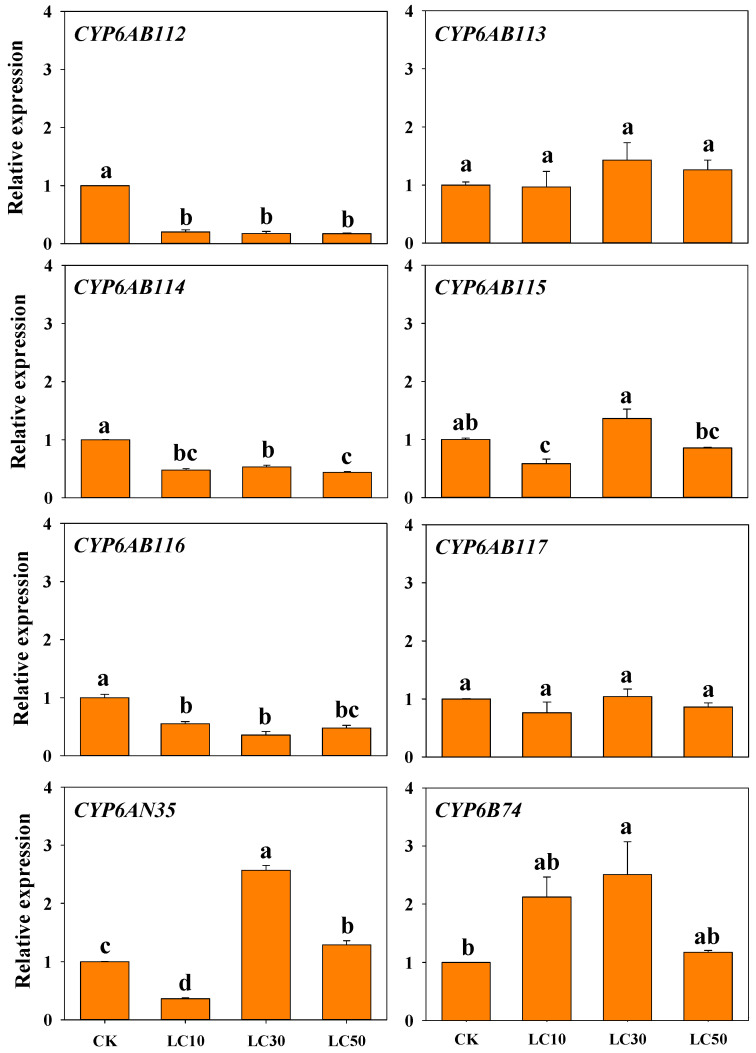
Relative expression of CYP6 genes in *G. molesta* treated with different concentrations of deltamethrin. Note: control (CK), lethal concentration 10% (LC_10_), lethal concentration 30% (LC_30_), and lethal concentration 50% (LC_50_). Each treatment included three biological replicates. The data in the bar charts are presented as the means ± SEs. Different letters indicate significant differences at the *p* < 0.05 level, as determined by ANOVA and Tukey’s HSD test.

**Figure 7 insects-13-00300-f007:**
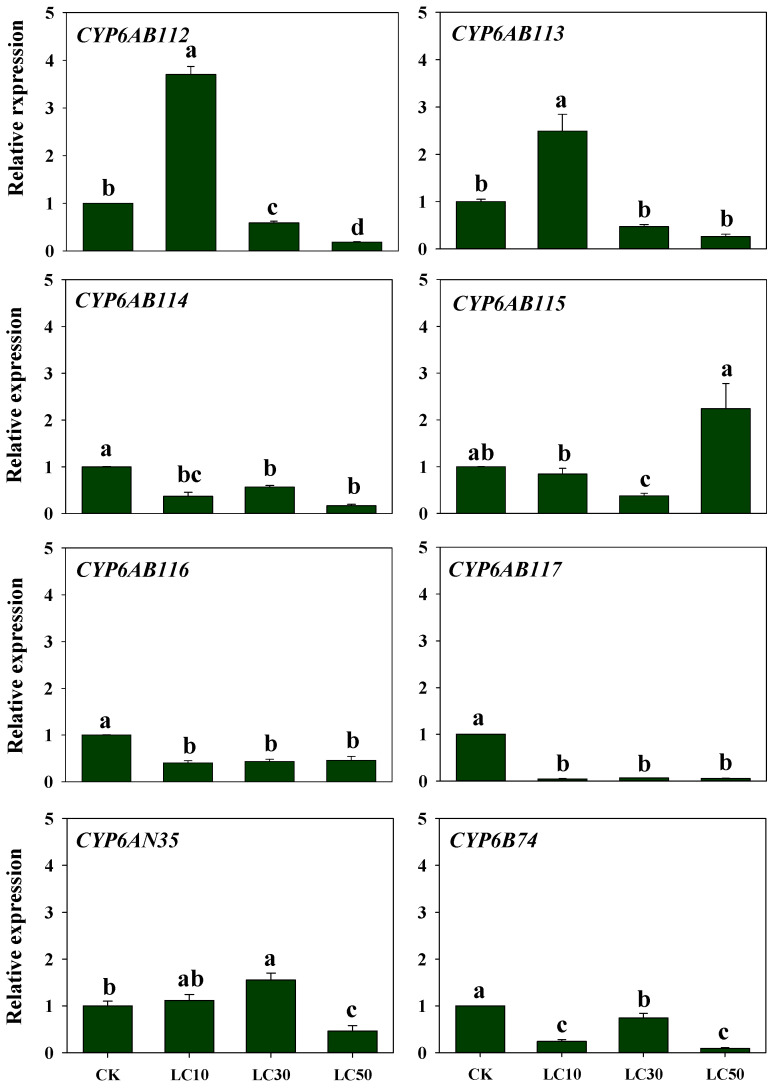
Relative expression of CYP6 genes in *G. molesta* treated with different concentrations of chlorantraniliprole. Note: control (CK), lethal concentration 10% (LC_10_), lethal concentration 30% (LC_30_), and lethal concentration 50% (LC_50_). Each treatment included three biological replicates. The data in the bar charts are presented as the means ± SEs. Different letters indicate significant differences at the *p* < 0.05 level, as determined by ANOVA and Tukey’s HSD test.

## Data Availability

The data presented in this study are available in article or Appendix A.

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
