# Peer review of "Identification and Characterization of CYP6 Family Genes from the Oriental Fruit Moth (Grapholita molesta) and Their Responses to Insecticides"

_insects, 2022, doi:10.3390/insects13030300_

Round 1
Reviewer 1 Report
In their manuscript: “Identification and characterization of CYP6 family genes from the oriental fruit moth, Grapholita molesta, and their responses to insecticides”, Hui Han et al. present a bioinformatics study that identified 8 genes of the CYP6 family of P450 enzymes. P450s are known to be involved in insecticide resistance. They further get the predicted 3D structure of the encoded CYP6s and they analyzed the differential expressions of these CYP6 genes during development of the insect and in its different tissues. This is therefore a comprehensive study but I have some minor comments that I would like to be taken into consideration by the authors.
Minor comments
- Figure 1 as such is too small; please provide an enlarged version in the manuscript.
- On the 3D structures shown on Fig.1, where is the N-terminal transmembrane segment typical of microsomal eukaryotic P450 enzymes? Why this segment is not apparent?
- Swiss-Model is a homology modeling algorithm with its well-known bias: the 3D structure you get is necessarily close to those of the template structures chosen. Since the summer of 2021, there is an artificial intelligence tool that allows highly accurate ab initio (i.e. without the need of templates) prediction of 3D structure from the amino acid sequence, this is AlphaFold2. Why did the authors not use this new, efficient tool instead of the old Swiss-Model approach?
- The particularity of CYP6AN35 being the only of these CYP6s new genes to be highly expressed in the head of the moth. The authors write that this is the sign of its implication in xenobiotic metabolism. But this is highly likely that function of all of these 8 CYP6 genes. So, this particularity of CYP6AN35 has to deal probably with a particular protection of head organs (I was thinking of the outh and the proboscis) against particular insecticides (since the expression of this P450 does not seem to be highly induced by chlorantraniliprole. Could the authors comment on that?
- On page 4, the expression “Meander domain” is not correct, it should be only “meander” (not stating domain) and with a lower case “M”.
- A very important review article is missing in the Reference list, namely: Xianchun Li, Mary A. Schuler, and May R. Berenbaum (2007) Molecular Mechanisms of Metabolic Resistance to Synthetic and Natural Xenobiotics. Annu. Rev. Entomol. 52, 231-253. It should be added to the reference list.
Reviewer 2 Report
Han et al. tested eight CYP6 gene expressions in various tissues and different stages of oriental fruit moth, as well as in the response to three types of insecticides. The authors tried to identify the potential function of these CYP6 genes in insecticide detoxification using insecticide induction and real-time PCR methods. Here are some comments below for the authors to improve this study.
- In the introduction, the authors may consider citing more references. Line 39, needs references to state that “the population of G. molesta……has become more and more severe.”, and lines 40-41, “the repeated use of ……the development of resistance…”. In line 82, the authors need to explain how the new control methods could be developed with more description and prediction.
- In the materials and methods, the authors mentioned that G. molesta were collected from the field. Have authors tested the initial insecticide susceptibility after collection? This question is based on the considering to the propose of this study focused on the cytochrome P450 genes in insecticide detoxification and the insects were never exposed to any insecticides for 5 years after collection.
The authors mentioned that the spermary and ovary were dissected but no description for the number of male and female moths.
The authors stated the “Bioinformatics analysis” for 8 CYP genes. How did the authors obtain the gene sequence information? Did authors conduct the transcriptome analysis, or just obtain the gene information from published research data? Either of these needs to be explained with detailed information.
The induction of expression method was only tested on adult moths, which didn’t indicate male or female moths either. Why only test insecticide induction in adult moths? Most of the CYP6 genes in this study are overexpressed in the larval stage and thus it is suggested to test the insecticide induction at the larval stage as well.
Based on the results of the insecticide induction test some of the CYP6 genes were not induced expression and authors concluded that because of a kind of inhibition of deltamethrin on CYP6AB112, -6AB114, and -6AB116 expression. It is not a precise conclusion, because some P450 genes can be induced expression before or after 24 hours exposure to insecticides, such as 12 hours, 48 hours, and 72 hours. However, the authors only tested the gene expression of CYP genes after 24 hours of exposure. It is suggested that the authors add the time point method along with the dose range for P450 gene induction.
For the reference gene of real-time PCR, the authors either need experiment-based evidence to the ᵦ-actin is consistent expressing or provide references.
Round 2
Reviewer 1 Report
The corrections introduced by the authors correspond to my comments and address them correctly.
Reviewer 2 Report
It is appreciated that the authors did a good job to revise the manuscript and respond the comments. However, here are a few comments for authors to revise the manuscript.
- Add reference(s) in "2.2 Bioinformatics analysis" since the RNA-seq and transcriptome analysis has been done.
- The authors need to carefully conclude the inhibition of CYP gene expression by insecticides because the authors did not screen enough time-points after insecticide treatment.
Author Response
Dear editor:
Thank you for your decision letter on our manuscript [Manuscript ID insects-1594612]. We appreciate your and the reviewers’ comments. We have revised the manuscript in accordance with the reviewers’ comments and carefully proofread the manuscript to eliminate typographical, grammatical, and bibliographical errors. To aid identification of the revisions, we used the “track changes” function. Our point-by-point responses to the reviewers’ comments are as follows:
Response to Reviewer 2 Comments
Point 1: Add reference(s) in "2.2 Bioinformatics analysis" since the RNA-seq and transcriptome analysis has been done.
Response 1: We thank the reviewer for the comments. As suggested by the reviewers, we have added the corresponding references to the text (lines 108, P3).
Point 2: The authors need to carefully conclude the inhibition of CYP gene expression by insecticides because the authors did not screen enough time-points after insecticide treatment.
Response 2: Thank you very much for your suggestion. We've added this part in the section of the Discussion (line 320-336, P13), The modified part is marked with green font.
Round 3
Reviewer 2 Report
The authors have modified the manuscript according to the comments.